

# The lifetimes and potential change in planetary albedo owing to the oxidation of organic films extracted from atmospheric aerosol by hydroyxl (OH) radical oxidation at the air-water interface of aerosol particles

Rosalie H. Shepherd[1,2], Martin D. King[1,*], Andrew D. Ward[2], Edward J. Stuckey[1,3], Rebecca J.L. Welbourn[3], Neil Brough[4,†], Adam Milsom[5], Christian Pfrang[5,6], and Thomas Arnold[7‡]

[1]Centre of Climate, Ocean and Atmosphere, Department of Earth Sciences, Royal Holloway University of London, Egham, Surrey, TW20 0EX, UK

[2]Central Laser Facility, Rutherford-Appleton Laboratory, Harwell Science Campus, Chilton, Didcot, Oxfordshire, OX11 0FA, UK

[3]ISIS Pulsed Neutron and Muon Source, Rutherford-Appleton Laboratory, Harwell Science Campus, Chilton, Didcot, Oxfordshire, OX11 0FA, UK

[4]British Antarctic Survey, Natural Environmental Research Council, High Cross, Madingly Road, Cambridge, CB3 0ET, UK

[†]Present address: National Institute of Water and Atmospheric Research, Wellington, New Zealand

[5]Geography, Earth and Environmental Sciences, Life and Environmental Sciences, University of Birmingham, Birmingham, B15 2TT, United Kingdom

[6]Department of Meteorology, University of Reading, Whiteknights, Earley Gate, Reading, RG6 6ET, UK

[7]Diamond Light Source, Harwell Science Campus, Chilton, Didcot, Oxfordshire, OX11 0DE, UK

[‡]Present address: European Spallation Source, Partikelgatan, 2224 84, Lund, Sweden

**Correspondence:** Martin D. King (m.king@rhul.ac.uk)

**Abstract.** Water insoluble organic material extracted from atmospheric aerosol samples collected in urban (Royal Holloway, University of London, UK) and remote (Halley, Antarctica) locations were shown to form stable thin films at an air-water interface, these organic films reacted quickly with gas-phase OH radicals which may impact planetary albedo. The x-ray reflectivity measurements additionally indicate that the film may be consistent with having a structure with increased electron density of

film molecules towards the water, suggesting amphiphilic behaviour. Bimolecular rate constants for gas-phase OH radical oxidation of urban or remote aerosol extracts were typically of the order $\sim 10^{10}$ cm$^3$molecule$^{-1}$s$^{-1}$, giving atmospheric lifetimes of the film with respect to gas-phase OH radical oxidation of minutes at typical atmospheric OH radical concentrations. Kinetic modelling of core-shell droplet dynamics suggests film lifetime of a few minutes, depending on ambient OH radical mixing ratio. Modelling the oxidation kinetics with KM SUB suggests half-lives of minutes to an hour and values of $k_{surf}$ of $\sim 2 \times 10^{-7}$

and $\sim 5 \times 10^{-5}$ cm$^2$ s$^{-1}$ for urban and remote aerosol film extracts respectively. The lifetimes and half-lives calculated at typical OH atmospheric ambient mixing ratios are smaller than the typical residence time of atmospheric aerosols and thus oxidation of organic material should be considered in atmospheric modelling. Thin organic films at the air-water interface of atmospheric aerosol or cloud droplets may alter the light scattering properties of the aerosol. X-ray reflectivity measurements of atmospheric aerosol film material at the air-water interface resulted in calculated film thickness values to be either $\sim 10$ Å or

$\sim 17$ Å for remote or urban aerosol extracts respectively and oxidation did not remove the films completely. One dimensional





radiative transfer-modelling suggest the oxidation of thin organic films on atmospheric particles by OH radicals may reduce the planetary albedo by a small, but potentially significant amount.

## 1 Introduction

Atmospheric aerosols significantly contribute to the Earth's climate (IPCC, 2021) and directly influence the proportion of solar energy reaching the Earth's surface by scattering or absorbing incoming solar radiation (IPCC, 2021), and indirectly through cloud condensation nuclei (Bréon et al., 2002; Lohmann and Feichter, 2005; Ramanathan et al., 2001; Haywood et al., 2023; Merikanto et al., 2009; Rosenfeld et al., 2008; Seinfeld et al., 2016). Current understanding of the effect of atmospheric aerosols on climate is considered low (IPCC, 2021; Burkholder et al., 2017; Prather et al., 2008). The complex chemical composition

and variable physical properties associated with atmospheric aerosols makes it difficult to obtain complete chemical information on aerosols (Jacobson et al., 2000), and is consequently a contributory factor to the uncertainty surrounding atmospheric aerosols. An atmospheric aerosol may be composed of several different substituents that may phase separate in the individual aerosol (Reid et al., 2011; Russell et al., 2002), often resulting in the formation of a thin organic film forming around a core aerosol droplet or particle (Donaldson and Vaida, 2006; Ellison et al., 1999; Gill et al., 1983; Tervahattu et al., 2002; Wyslouzil

et al., 2006; Jones et al., 2015; Kirpes et al., 2019; Jones et al., 2017; Barker et al., 2023), a so called core-shell aerosol. The presence of a thin organic film may alter the chemical and physical properties of the core-shell aerosol through (a) altering the transport of chemicals from the gas to liquid phase and vice versa (Donaldson and Anderson, 1999; Enami et al., 2014a), (b) reducing the rate of evaporation from the core aerosol (Davies et al., 2013; Eliason et al., 2003; Kaiser et al., 1996; McFiggans et al., 2006), (c) altering the cloud condensation nuclei activation potential, (Cruz and Pandis, 1998; Ruehl and Wilson, 2014;

King et al., 2009; Ovadnevaite et al., 2017), (d) reducing the scavenging of the core aerosol by larger cloud and ice particles (Andreae and Rosenfeld, 2008; Feingold and Chuang, 2002) and (e) changing the aerosol optical properties and thus the light scattering properties of the aerosol (Jones et al., 2015; Shepherd et al., 2022; Barker et al., 2023).

The organic-coated aerosol is susceptible to atmospheric oxidation (Eliason et al., 2003; Mmereki and Donaldson, 2003;

Jones et al., 2017, 2023; Milsom et al., 2022b; Shepherd et al., 2022; King et al., 2009, 2020) and as a result, may change both chemically and physically (King et al., 2004). Many oxidant species exist in the atmosphere and hydroxyl, OH, radicals are one of the dominant oxidising chemicals in the lower atmosphere (Prinn et al., 2001). There has been recent evidence that OH radicals are produced spontaneously in the dark at the surface of aqueous droplets (Li et al., 2023). Understanding the behaviour of organic aerosol upon exposure to OH radicals is paramount for determining the atmospheric chemical lifetime of

the aerosol and subsequently a number of studies have focused on the oxidation of atmospheric aerosols when exposed to OH radicals. These studies include the oxidation between sub-micron aqueous aerosols when exposed to OH radicals, (McNeill





et al., 2007), the heterogeneous reaction between OH radicals and sub-micron unsaturated fatty acid particles (Nah et al., 2013), the oxidation of alkanoic acids at the air-water interface by OH radicals (Enami et al., 2014b), and the effect of sulphur dioxide on the oxidation rate of organic aerosol by OH radicals (Richards-Henderson et al., 2016). A more exhaustive list of reaction

of gas-phase OH radicals with surface bound organic compounds can be found in table 3 of Shepherd et al. (2022).

A proxy for the interface of organic coated aqueous aerosol can be produced by the use of thin films of proxy atmospheric material at the air-water interface, a method which has been implemented extensively (Chapleski et al., 2016; Cosman et al., 2008; Dennis-Smither et al., 2012; Eliason et al., 2004; George et al., 2007; González-Labrada et al., 2006, 2007; Jones et al.,

2023; King et al., 2009, 2020; Knopf et al., 2007; Pfrang et al., 2014; Nakayama et al., 2013; Slade and Knopf, 2014; Sebastiani et al., 2018, 2022; Vieceli et al., 2004; Voss et al., 2006; Woden et al., 2021; King et al., 2020; Shepherd et al., 2022; Jones et al., 2017, 2023). One approach to measuring the structure of a thin film at the air-water interface is x-ray and neutron reflectometry, which enables the physical characterisation of a thin film at Ångström resolution, including observing structural changes to the interface in real time. A number of studies that focused on organic films at the air-water interface have employed

reflection techniques, for example the oxidation of a film of oleic acid at the air-water interface with gas-phase ozone (King et al., 2009, 2020) or gas-phase nitrogen dioxide (King et al., 2010), the oxidation of methyl oleate thin films with gas-phase ozone (Pfrang et al., 2014), changes owing to the chemical denaturation of $\beta$-lectoglobulin (Perriman et al., 2007), the structure of mixed monolayers with varying concentration of phospholipids and lipids with polymer headgroups (Majewski et al., 1997), adsorption behaviours of DNA and sodium poly(styrene sulfonate) when exposed to cationic gemini surfactants (Vongsetskul

et al., 2009), the self assembly of conjugated polymers at the air-water interface (Reitzel et al., 2000), the formation peptide sheets from the denaturation of proteins at the air-water interface (Gidalevitz et al., 1999), and the oxidation of lung lining (Hemming et al., 2015, 2022; Thompson et al., 2013, 2010).

Other techniques have also been successfully applied to study thin films at the air-water interface: González-Labrada et al. (2006) recorded the change in surface tension of a film composed of oleic acid when exposed to ozone using a surface tensiome-

ter, whilst Wadia et al. (2000) studied unsaturated and saturated phospholipids by using an atmospheric pressure ionisation mass spectrometer.

Investigations using atmospheric aerosol extracted from the atmosphere are critical to provide insight into the atmospheric process and composition compared to proxies, however these studies are sparse. Examples of such studies include Zhou et al. (2014) who studied the heterogeneous oxidation of seawater polyunsaturated fatty acids at the air water interface; a study

relevant to aerosol research owing to the material at the surface of the ocean becoming incorporated into atmospheric aerosol (Blanchard, 1964; Donaldson and George, 2012; Marty and Saliot, 1976), whilst Jones et al. (2017) applied X-ray reflection techniques to study the oxidation of thin films extracted from atmospheric aerosol extracts collected at Royal Holloway, University of London and seawater samples collected from the English Channel by aqueous-phase OH oxidation and gas-phase ozone oxidation. Shepherd et al. (2022) exposed thin films of atmospheric aerosol extracts collected from Royal Holloway,

Antarctica and wood smoke burning to OH radicals and estimated the atmospheric chemical half-life of such films based on a



KM SUB kinetic model analysis (Shiraiwa et al., 2010).

The purpose of the study described here is four-fold: firstly to confirm that organic films extracted from atmospheric aerosol can form stable films at the air-water interface, secondly, determine the film thickness for organic matter extracted from atmo-
spheric aerosol at the air-water interface, thirdly to determine the chemical lifetime of a film of atmospheric aerosol extract at the air-water interface, and lastly to estimate the relevance of the oxidation of organic films on aqueous and mineral aerosol to planetary albedo. The study presented here will yield morphological data about the film at the air-water interface through calculating film thickness to Angstrom precision: accurate estimation of film thickness is paramount to model the light scattering from particles with films likely to be found in the atmosphere owing to the importance of the film thickness in the ability
of core-shell aerosol to heat or cool the atmosphere (Barker et al., 2023). The chemical lifetime, with respect to gas-phase OH radical will be estimated by applying a kinetic model (KM Sub) to the experimental data, extrapolating to a range of atmospherically relevant OH radical concentrations. Lastly the thickness of the organic film before and after oxidation by OH radical will be used to estimate the the top of the atmosphere albedo change owing to chemical oxidation by OH radical for a series of exemplar aerosol distributions.

## 95   2   Method

X-ray reflectivity was applied to study a thin film of atmospheric aerosol at the air-water interface upon exposure to gas-phase OH radicals; the following section very briefly describes the X-ray reflectivity technique and the kinetic model employed to determine the chemical lifetime of these films. Additionally, the method applied to collect the insoluble, surface-active atmospheric aerosol extracts for use on the x-ray beamline is described.

## 100   2.1   Extraction of atmospheric aerosol

Organic material from atmospheric aerosols was extracted from quartz fibre filters collected at two locations: the campus of Royal Holloway, University of London during the months of September 2015 and January 2016, and at the Halley Clean Air Sector Laboratory operated by the British Antarctic Survey over the Antarctic summers of 2015 and 2016. The aerosol extract collected from Royal Holloway, University of London has been classified as urban owing to the proximity of the city of Lon-
don, major motorways such as the M25, M3 and M4, and the international airport, Heathrow all lying within a radius of 30 km. In contrast, because of the physical remoteness Antarctica is considered a clean environment, and therefore for the purpose of the study Antarctic aerosol extract is described as remote. The urban samples were each collected over an approximately 30-day period, whilst the remote samples were collected over approximately 60 days.

For the urban sampling sites, the aerosol was extracted from the atmosphere through an air pump that pulled air through clean stainless-steel pipelines at a flow of 30 L min$^{-1}$, whilst remote aerosol was sampled by pulling air through a short length (10 cm)of quarter inch O.D. Perfluoroalkoxy, (PFA) Teflon tubing by applying a low volume air sampler (Model VM-4) with





a flow rate of 20 L min$^{-1}$. In both sampling sites, the pipelines led to a PFA Savillex commercial filter holder, in which a pre-combusted 47 mm diameter quartz filter was encased. Atmospheric aerosol extract was collected upon the filter. To separate

the insoluble organic material from the filter, the filter was shaken manually and very gently sonicated for five minutes in a solution consisting of 10 ml ultra-pure water ($< 18$ MΩcm) and 10 ml chloroform (Sigma-Aldrich, 0.5-1 % ethanol as stabilizer); shaking commonly reduced the filter to a pulp. The pulp was subsequently filtered with another pre-combusted quartz filter to remove the filter paper: the pulp was washed with ultra-pure water and chloroform several times to maximise the amount of sample extracted. The sonication was not found to be detrimental to the material being extracted (Jones et al.,

2017). The resulting filtrate consisted of a chloroform layer in which the material likely to form insoluble films at an air-water interface resided (Jones et al., 2008) and an aqueous layer. The chloroform layer was separated from the aqueous layer by passing the filtrate through a separating funnel. The chloroform was then removed from the atmospheric aerosol extract by evaporative blow-down leaving an oily residue behind. The residual material was stored in 2 ml of fresh chloroform at $-18°$C in the dark until use on the beam line. All glassware used in the extraction process was thoroughly cleaned with ultrapure

water and chloroform prior to use and the extraction process was carried out in a clean glove bag. Analytical blanks were filters extracted in the same manner as filters used to collect atmospheric aerosol. Travelling filters were analytical blanks that travelled to and from the collection sites under the same conditions as the filters used to collect atmospheric aerosol.

Analytical and travelling filter blanks were likewise extracted in exactly the same manner. The analytical blanks provided a method of determining the level of contamination resulting from the extraction process, whilst for the remote samples the

amount of contamination resulting from travelling to Antarctica and back was determined from a travelling filter blank that travelled with the sample.

## 2.2   Experimental Set-up

The water was held using a PTFE trough (Ball et al., 2013), with a volume of $\sim 90$ cm$^3$, and surface area of $\sim 175$ cm$^2$, mounted onto an anti-vibration table. Owing to the small size and geometry of the trough in relation to the x-ray beam footprint

on the water surface, the trough was used without barriers or a surface tension sensor. The small size was a compromise between the x-ray beam footprint and the limited sample. A UV lamp containing two fluorescent germicidal lamps, each with an output wavelength centered on 254 nm, and were suspended above the trough. To encase the experimental environment and to allow the production of an atmosphere containing OH radicals, the trough and UV lamp were enclosed within a Tedlar bag; two thin Kapton windows at either end of the trough facilitated travel of the x-ray beam into and out of the enclosed environment. The

Tedlar bag had a stainless-steel inlet and an exhaust to allow gas to enter and leave. The bag had a 25 L volume and a surface area of 612 cm$^2$. Generation of gas-phase OH radical required a saturated atmosphere of water vapour. To ensure the relative humidity of the experiment environment was maintained, a water reservoir with an approximate volume of $\sim 50$ cm$^3$ was included within the Tedlar bag. For each experimental run the trough was cleaned with chloroform and ultra-pure water before a fresh 90 cm$^3$ volume of ultra pure water ($< 18$MΩcm) was poured into the trough. Between $100 - 200$ $\mu$l of atmospheric

aerosol extract in chloroform was added to the air-water interface using a Hamilton syringe.





## 2.3 X-ray reflectivity

The surface specular x-ray reflectometer at Diamond Light Source, I07, was applied to study the film of atmospheric aerosol extract at the air-water interface. I07 is described in detail by Nicklin et al. (2016) and Arnold et al. (2012) and the x-ray reflectivity technique is described in detail by Chateigner (2013), however, a brief description will be given here. The x-ray 150 beam had a wavelength of 0.992 Å, and was specularly reflected off the interface and onto a Pilatus detector, where the beam was measured as a function of momentum transfer, Q, defined as:

$$Q = \frac{4\pi \sin(\theta)}{\lambda} \tag{1}$$

versus specular reflectivity, R, which is defined as

$$R = \frac{16\pi^2}{Q^4} (2\rho)^2 \sin\left(\frac{Q\delta}{2}\right)^2 \tag{2}$$

from Equation 1, $\lambda$ is defined as the wavelength of the specular x-ray and $\theta$ the incident angle of the specular x-ray beam onto the interface of study, whilst $\delta$ is the film thickness and $\rho$ the x-ray scattering length density. The x-ray scattering length density is related to the electron density of the interface, therefore heavier atoms that have a greater number of electrons will have a larger electron density and thus a greater x-ray scattering length density (Jones et al., 2017). The profile of momentum transfer versus reflectivity will be called an x-ray reflection profile henceforth in the paper. To optimise signal relative to time, 160 x-ray reflection profiles were collected over a time period of eight minutes. To determine precise values of x-ray scattering length density and film thickness, the experimentally collected x-ray reflection profiles were simulated by an Abelès formalism (Abelès, 1950) in the computer software Motofit(Nelson, 2006). The Abelès formalism used a model of the atmospheric aerosol extract at the air-water interface, including parameters for the x-ray scattering length density, film thickness and film roughness to simulate the experimentally determined x-ray reflectivity profile. The values were initially estimated and then refined until 165 a fit between experimental, $R^{\text{Experimental}}(Q)$, and simulated, $R^{\text{Calculated}}(Q)$, data was determined: application of a fitting metric $\chi^2$ ensured the final value provided the best fit where

$$\chi^2 = \frac{\left(R^{\text{Calculated}}(Q) - R^{\text{Experimental}}(Q)\right)}{R^{\text{Calculated}}(Q)} \tag{3}$$

Typically for a film of known composition at an air-water interface the surface coverage (number of molecules per unit area) can be calculated by $\frac{\rho\delta}{b}$, where b is the scattering length of a molecule. However in the study presented here the organic film is 170 (a) a complex mixture of unknown materials and (b) a reaction system producing unknown surface-active product molecules at the air-water interface whose relative amounts change with time. Thus the value of the scattering length, $b$, of the molecules in the film is not known. For the atmospheric aerosol extracts films exposed to gas-phase OH radicals, scattering length density per unit area, $\rho\delta$, was plotted as a function of time (Jones et al., 2017) instead. Such plots allowed the change in the film to be



monitored and for the quantity $\rho\delta$ to be used as a kinetic variable (Jones et al., 2017). Each point on the plots corresponds to a
new x-ray reflection profile; continuous collection of subsequent profiles allowed a temporal graph of $\rho\delta$ as a function of time
to be plotted. In contrast to the preliminary study by Jones et al. (2017) the study described here split the organic layer into two
separate layers with differing scattering length densities and thicknesses to better reproduce the experimental x-ray reflection
profiles. Upgrades in the beamline allowed this refinement of the reflectivity profile since the work of Jones et al. (2017). In
the study presented here the quantity, $\frac{\rho_t \delta_t}{\rho_{t=0} \delta_{t=0}}$, was followed as a function of time, $t$.

## 2.4  Gas-phase OH radical generation

Gas-phase OH radicals were formed from the photolysis of ozone in the presence of water vapour; oxygen saturated with
water vapour was flowed at $1$ L min$^{-1}$ through a photolytic ozoniser (Ultra-Violet Products Ltd) to produce ozone at a mixing
ratio of 0.85 ppm ( $2.1 \times 10^{13}$ molecule cm$^{-3}$). The oxygen was saturated with water vapour by bubbling through water. The
gas-phase concentration of OH radical in the Tedlar bag was estimated to be $6 \times 10^6$ molecule cm$^{-3}$. The concentration of OH
radical was calculated by kinetically modelling (*i.e.* solution of first-order differential equations using a Runge-Kutta Solver
(Press et al., 1987)) of reactions 1–30 in Atkinson et al. (2004) which cover the basic HOx and Ox reactions occurring in the
photolysis of ozone in the presence of water vapour. A first-order wall loss of OH radicals, $2$ s$^{-1}$, was added using the method
outlined in Dilbeck and Finlayson-Pitts (2013) based on a 25L Tedlar bag with a surface area of $0.612$ m$^2$ and assuming the
reaction probability for OH radicals on Tedlar will be similar to halocarbon wax, $\gamma = 6 \times 10^{-4}$ (Bertram et al., 2001). The
photolysis rate constants for the photolysis of ozone, $J\left(\mathrm{O}(^1D)\right)$ and hydrogen peroxide, were determined with an intensity
calibrated Metcon radiometer (normally used for measurements of atmospheric photolysis reactions (e.g. Kukui et al., 2014))
and the value of the rate constants for the photolysis of molecular oxygen was the value for ozone scaled by the product of
the absorption cross-sections and the quantum yields. The concentration of ozone within the bag was measured by UV-VIS
spectrometry after sampling in a 10 cm glass cell. The concentration of water vapour was based on the vapour pressure of water
(Haynes, 2016) to be 2.34 kPa at 20°C.

The reflectivity of an atmospheric aerosol extract at the air-water interface was recorded three times prior to oxidation to
establish that (a) the film was initially stable at the air-water interface and (b) to provide good signal to noise x-ray reflection
profiles to allow precise determination of x-ray scattering length density and film thickness. After the three measurements,
the film was exposed to gas-phase OH radicals. Continuous collection of x-ray reflection profiles during the exposure of the
atmospheric aerosol extract to gas-phase OH radicals allowed the change in the film to be analysed as a function of time.

## 2.5  Kinetic analysis

A central aim of the project was to estimate the atmospheric lifetime, $\tau$, of the atmospheric aerosol extract upon exposure to
gas-phase OH radicals. The rate of loss of atmospheric aerosol extract

$$\text{Film + OH} \rightarrow \text{products} \tag{4}$$



with time was described by:

$$\frac{d\Gamma_{\text{film}}}{dt} = k_4 \, [\text{OH}] \, \Gamma_{\text{film}} \tag{5}$$

where $\Gamma_{\text{film}}$ represents the surface coverage of the film, at the air-water interface, [OH], the concentration of gas-phase OH radicals and $k_4$ the bimolecular rate constant for reaction 4. Solving equation 5 gives:

$$\frac{\Gamma_{\text{film}}^{\text{t}}}{\Gamma_{\text{film}}^{\text{t=0}}} = e^{-k't} \tag{6}$$

where $k' = k_4 [\text{OH}]$ and substituting $\frac{\rho_{\text{t}} \delta_{\text{t}}}{\rho_{\text{t=0}} \delta_{\text{t=0}}}$ gives

$$\frac{\rho_{\text{t}} \delta_{\text{t}}}{\rho_{\text{t=0}} \delta_{\text{t=0}}} = e^{-k't} \tag{7}$$

the relative change in organic film surface coverage can be represented by the relative change in the measured scattering length per unit area, $\frac{\rho_{\text{t}} \delta_{\text{t}}}{\rho_{\text{t=0}} \delta_{\text{t=0}}}$. In Equations 5 and 6, $t$ represents time and $k'$ the pseudo-first order rate constant (equal to $k_4 [\text{OH}]$). The value of $k'$ was determined by plotting $\frac{\rho_{\text{t}} \delta_{\text{t}}}{\rho_{\text{t=0}} \delta_{\text{t=0}}}$ as a function of time, $t$, and fitting the subsequent plot to an exponential

decay, $e^{-k't}$.

The atmospheric lifetime, $\tau$, of the film with respect to oxidation by gas-phase OH radicals in the atmosphere can be calculated from the values of the rate constant determined for reaction 4, $k_4$, and a typical gas-phase concentrations of OH radicals in the atmosphere, $[\text{OH}]_{\text{atm}}$,

$$\tau = \frac{1}{k_4 [\text{OH}]_{\text{atm}}} \tag{8}$$

Three ozone-only control kinetic runs were performed in the absence of UV light to demonstrate no apparent loss of film material owing to reaction with gas-phase ozone (as opposed to the hydroxyl radical). A control kinetic run with oxygen only (no ozone or UV light present) was not required as no loss of film was observed with ozone/oxygen mixtures. A blank run with UV illumination only (i.e. no ozone) is not reported as the UV lamps generate ozone and subsequently OH radicals (Shepherd et al., 2022) from the photolysis of molecular oxygen.

**2.6   Kinetic modelling of the film-OH radical reaction**

The reaction system was modelled using the kinetic model of aerosol surface and bulk chemistry (KM-SUB) (Shiraiwa et al., 2010), which based on the Pöschl-Rudich-Ammann framework (Pöschl et al., 2007). OH radical surface adsorption and desorption are resolved along with the surface reaction between OH radicals and the insoluble film material. The inverse square of the film thickness was used to estimate the average initial surface concentration of film molecules, assuming the film thickness



is the average molecular length from the x-ray reflectivity measurements. The KM-SUB model constructed here is analogous to previous modelling work reactions with insoluble films (Shepherd et al., 2022), which employs the reaction scheme presented in equation 4 with the addition of a residual film that may be present. A global optimisation algorithm (differential evolution) (Storn and Price, 1997) was used to optimise the model, with only the surface reaction coefficient ($k_{\mathrm{surf}}$) varied. Other parameters were held constant (see the model described in our previous study on aerosol extract oxidation kinetics (Shepherd et al.,

2022)). The Langmuir-Hinshelwood surface reaction mechanism is assumed due to the evidence that OH radicals favour this mechanism (Arangio et al., 2015; Bagot et al., 2008; Enami et al., 2014b). The uncertainty in the value of $k_{\mathrm{surf}}$ was estimated using a Markov Chain Monte Carlo (MCMC) algorithm (Foreman-Mackey et al., 2013; Hogg and Foreman-Mackey, 2018). A description of the model and details of MCMC sampling procedure are presented in a previous study on a similar system (Shepherd et al., 2022). The models were constructed and optimised using MultilayerPy, a framework for building kinetic mul-

tilayer models (Milsom et al., 2022a). Jupyter notebooks along with the model code are available alongside this publication (see code availability).

### 2.7   1-d Radiative Transfer and Mie modelling

A combination of Mie light scattering modelling and 1-D radiative transfer modelling was undertaken to quantify the effects of the oxidation-caused change in film thicknesses observed in this work on the top of atmosphere albedo. The models used

were adapted from Bohren and Huffman's core-shell model 'bhcoat' (Bohern and Huffman, 1998), and the TUV (Tropospheric Ultraviolet-Visible) radiation transfer model (Madronich and Flocke, 1999). The core-shell Mie model was used to produce the wavelength-dependent single-scattering albedos, and the asymmetry parameter of mineral spheres coated in organics averaged over different aerosol population. The values of the real refractive indexes of silica and water were taken from Kitamura et al. (2007), and Schiebener et al. (1990) respectively, and the values of the imaginary refractive indexes were taken from Khashan

and Nassif (2001) and Hale and Querry (1973). The temperature of silica and water was assumed to be 293K for calculating optical properties. Values of the complex refractive indexes of urban and remote organic thin film material were taken as a combination of real components produced by Shepherd et al. (2018) and imaginary components by Kirchstetter et al. (2004) and Virkkula et al. (2022). Calculations of the value of single scattering albedo and asymmetry parameter were calculated over a range of core sizes of 10–1000 nm using the characteristic size distribution functions of aerosol from the respective

environments as published by Jaenicke (1993). The calculation was repeated for film thicknesses between 0.05 nm to 50 nm. The wavelength-dependent single-scattering albedo (SSA) and the asymmetry parameter were used in an atmospheric 1-d radiative transfer model to estimate the impact of the oxidation of films on top of atmosphere albedo. The TUV model was used with minimal adjustments. Three consecutive 1km layers were placed on the surface to form a 3km thick planetary boundary layer of aerosol, which were given the optical properties generated from Mie modelling with an aerosol optical depth

for each layer of 0.235. The rest of the atmosphere was described by an Elterman aerosol profile with no clouds present. The top of atmosphere albedo was calculated as a ratio of upwelling and downwelling irradiance at an altitude of 80 km with a solar zenith angle of $60°$. The change in top of the atmosphere albedo owing to the oxidation of organic films, ΔAlbedo was determined by subtracting the top of atmosphere albedo calculated using aerosol with a given film thickness from that of a





model using the same aerosol with films of half that thickness. The changes owing to oxidation for single scattering albedo,
$\Delta$SSA, and the asymmetrical light scattering, $\Delta g_{\mathrm{sca}}$, were also produced. Four profiles of $\Delta$Albedo, $\Delta g_{\mathrm{sca}}$ and $\Delta$SSA versus
initial film thickness were produced which were representative of organic coated silica and water aerosol whose size and film
optical properties were representative of urban and remote environments from Jaenicke (1993).

## 3 Results and Discussion

The results and discussion of the x-ray reflectivity of thin films of material extracted from atmospheric aerosol at the air-water
interface are presented, followed by a kinetic analysis of its oxidation with gas-phase OH radical.

### 3.1 X-ray reflection and properties of the thin film

Organic thin films composed of material extracted from the atmospheric aerosol were studied at the air-water interface. Fig. 1
depicts the x-ray reflection profiles for aerosol extracts sourced from (a) urban and (b) remote locations along with the cor-
responding analytical (or travelling) blank and clean air-water interface. The figure demonstrates a definite change in the
reflectivity profile with and without the atmospheric aerosol extract sample. The figure also demonstrates the travelling and
analytical blanks are effectively indistinguishable from the clean air-water interface. Very close inspection of Fig. 1B suggests
the traveling blank may have received a tiny quantity of contamination on its long return journey from Antarctica. However,
the changes in the x-ray reflectivity profile relative to the clean air-water interface is many orders of magnitude smaller than
the sample. The agreement is remarkable. Fig. 1 demonstrates the essential requirement for these experiments to be performed
at x-ray synchrotrons as the difference between sample and no sample at the air-water interface requires a large signal to noise
ratio in reflectivity at large value of Q, momentum transfer values. A commercial x-ray reflectometer is generally unlikely to
achieve sufficient signal-to-noise ratios at the larger values of Q (plotted in Fig. 1) in timescales important for studying kinetics.

Application of Abelès formalism allowed the x-ray scattering length density and film thickness of the aerosol extract film to
be calculated. Additionally, more detailed structural information of the extract at the air-water interface was determined; the
urban and remote aerosol extracts both demonstrated a good fit between experimental and calculated x-ray reflectivity profile
to a two-layer system (as opposed to a one or three-layer system). It was not possible to achieve a realistic reproduction of the
experimental reflectivity profile with one layer at the air-water interface and although a three -layer system provided a good
visual fit it did not lower the value of the fitting metric, $\chi^2$. The two-layer system may be indicative of a film that has a structure
with increased electron density of film molecules towards the water, suggesting amphiphilic, lipid-like, behaviour.

The layers of the two-layer system will be described as a layer beside the air, "air interface" and a layer at the water interface,
"water interface". The x-ray scattering length densities and film thickness determined for urban and remote atmospheric aerosol
extracts are displayed in Table 1. Uncertainty in the values displayed in Table 1 was determined by adjusting the scattering
length density or film thickness in turn until the simulated data no longer resembled the experimental data, as determined by



eye. Application of a $\chi^2$ test (see equation 3) ensured the x-ray scattering length density and film thickness values displayed were the values that gave the best fit. A demonstration of the robustness of the fitting procedure is given in Fig. 2. Fig. 2 is a plot of the value of the fitting metric, $\chi^2$, presented as a function of the x-ray scattering length density for three different film thickness. Fig. 2 demonstrates a well defined almost symmetrical minima. Note the minima of these curves is independent of

thickness near the "best fit". The upper panel is for layer of aerosol extract at air interface and the lower panel is for the layer at the water interface. Each layer was investigated independently and the data presented in Fig. 3 is from the urban sample. Fig. 6 suggests the x-ray scattering length density can easily be fitted to $\pm 0.05 \times 10^{-6}$ Å$^{-2}$.

    The number of samples studied is small owing to limited access to the x-ray synchrotron source and limited sample from

atmospheric aerosol (the Antarctic aerosol sample was completely consumed in these studies). However, inspection of table 1 demonstrates the thickness and scattering length density are consistent between samples from the same locale, but different between locales. The remote sample is indicative of background polar/marine aerosol (Wolff, 1990) and is a thinner film with a smaller electron density than the urban polluted sample. The urban polluted sample is characterised by its close proximity to the London mega city, a major international airport (Heathrow) and the junction of three major arterial motorways. Shepherd

et al. (2022) studied the oxidation of very similar organic films extracted from the atmosphere at the air-water interface in their work but used neutron reflectometry instead of x-ray reflectometry. The Antarctic samples were the same as used here, but the urban samples were the same location and year but collected in different months, A comparison is shown in table 3. The thickness were broadly comparable, the films were 10-20 Å thick. The film thickness is sensitive to the amount of materials successfully added to the air-water interface and more material was added in the x-ray work relative to the neutron work. Note

not all material attempted to be added to an air-water interface is successfully placed there and the concentrations and chemical identities of the materials are unknown.

    The separation of the atmospheric aerosol extract into two layers indicates that the extract may contain hydrophobic and hydrophilic components. For both urban and remote extracts, the x-ray scattering length density and film thickness for the

layer at the water interface is larger than the x-ray scattering length density for the layer at the air interface. Literature reports similar trends in x-ray scattering length density of thin films composed of molecules containing hydrophobic and hydrophilic regions, for example Dabkowska et al. (2013) studied monolayer films of the lipid distearoylphosphatidylcholine (DSPC) at the air-water interface and report a x-ray scattering length density larger by $5.55 \times 10^{-6}$ Å$^{-2}$ for the hydrophilic head group region than for the hydrophobic chain region of the lipid.


    Previous neutron and x-ray studies have used a film thickness ranging from $10-20$ Å for a thin film at the air-water interface (Jones et al., 2017; King et al., 2009; Pfrang et al., 2014) and 24–51 nm at the air-solid interface Milsom et al. (2022b). The air-solid films are spin-coated onto the solid surface. In the study presented, the total thickness of the atmospheric aerosol extract films did not exceed 18 Å for urban aerosol extracts or 10 Å for remote aerosol extracts, agreeing with previous mea-



surements. These thicknesses are important for atmospheric and core-shell morphology atmospheric modelling (Jones et al.,
2015; Shepherd et al., 2022; McGrory et al., 2022) and section 2.7.

## 3.2  Oxidation of Atmospheric Aerosol

The films of atmospheric aerosol extract at the air-water interface were exposed to gas-phase OH radicals with a concentration
of $3.3 \times 10^6$ molecule cm$^{-3}$. The change in x-ray reflectivity was followed continuously in eight-minute intervals. Fig. 3 depicts

the change in the x-ray reflection profile with time as the atmospheric aerosol extract at the air-water interface was exposed to
gas-phase OH radicals. Fig. 3 clearly shows changes to the interface during oxidation and that oxidation does not completely
remove all material from the the interface back to just water. Results from the neutron reflectivity study of similar samples,
(Shepherd et al., 2022) also demonstrated some film remaining after oxidation, although perhaps less than shown in the study
presented here.

Each x-ray reflection profile collected was simulated and fitted with a calculated reflection profile to determine the x-ray
scattering length density and film thickness as a function of time. The experimental x-ray reflectivity profiles of the reacting
system were calculated and fitted as a function of time as one and two layer systems. The two-layer calculation of the x-ray
reflectivity profile gave a superior fit to the experimental data throughout the oxidation reaction. The values of the scattering
length density, $\rho$, and thickness, $\delta$, are plotted as $\frac{\rho_t \delta_t}{\rho_{t=0} \delta_{t=0}}$ versus time in Fig. 4. Figure 4 depicts the kinetic decay observed

for the films composed of urban aerosol extract upon exposure to gas-phase OH radicals, whilst Fig. 5 depicts the kinetic
decay observed for the films composed of remote aerosol extracts upon exposure to gas-phase OH radicals. Uncertainty bars in
Fig.s 3, 4 and 5 represent the propagated uncertainty (Bevington et al., 1993) from the standard deviations in x -ray scattering
length density and film thickness from data fitting.

Both Figs. 4 and 5 demonstrate an exponential decay in the scattering length per unit area, $\frac{\rho_t \delta_t}{\rho_{t=0} \delta_{t=0}}$ when each thin film was
exposed to OH radicals and it can be observed that the rate of decay for the two layers (irrespective of sample) lies within
error of each other. The decay in $\frac{\rho_t \delta_t}{\rho_{t=0} \delta_{t=0}}$ as a function of time shows that the interface is becoming more like pure water upon
exposure to OH radicals, suggesting that either the interface is becoming hydrated or material is being lost from the interface,
to water sub-phase of gas-phase owing to reactions with OH radicals. Figures 4 and 5 both demonstrate that urban and remote

aerosol extracts do not change when exposed to ozone only, therefore the reaction observed is consistent with reaction with
gas-phase OH radical. Jones et al. (2017) also observed no reactivity to gas-phase ozone when films of seawater extracts and
atmospheric aerosol extracts were also exposed to ozone.

To calculate the value of the bimolecular rate constant, $k_4$, for reaction 4 and because the kinetic decay profiles, $\frac{\rho_t \delta_t}{\rho_{t=0} \delta_{t=0}}$ vs

time were the same the value of $\rho \delta$ for the two layers were averaged and the decay fitted to an exponential curve of the form
$e^{-kt} + c$. Where, $k$, is a rate constant, $t$ is the reaction time and c represents the material remaining at the interface. Fig. 6
depicts the kinetic decay profiles for the reaction of the two urban and two remote atmospheric aerosol extracts with gas-phase
OH radical. The pseudo first-order rate constants calculated from figure 6 are displayed in Table 2 alongside the modelled





concentration of gas-phase OH radicals above the film. The bimolecular rate constant, $k_4$, for reaction 4 is calculated from

$k = k'[\text{OH}]$. Table 3 demonstrates that the values of the bimolecular rate constant for reaction 4 determined here in the study presented here and in the neutron reflection study Shepherd et al. (2022) are broadly similar with the exception of one very fast measurement presented here. Examination of the data used to determine the values of the rate constants in Fig. 5, panel a demonstrates that more temporal resolution may be needed to have confidence in this value. All the values of the bimolecular rate constants considered here are close to the diffusion limit.

### 3.3   Kinetic KM SUB Model Results

Values of $k_{\text{surf}}$ are larger for the remote samples compared with those of the urban samples, (table 2) The low number of data-points fitted to for both remote aerosol extract in the decaying portion of the decay curves reduces the precision of the model fit (Fig. 7). However, the general trend is consistent with the trend in $k'$ from the analytical fits to the decays (Table 2). Note that the KM SUB kinetic model also accounted for the initial average surface concentration of film molecules, estimated as

the inverse square of the initial film thickness, resulting in different initial surface concentrations of film molecules for each model run and the model was sensitive to this. Fitted values of $k_{\text{surf}}$ for reaction of the OH radical with the the remote samples are generally larger in the study presented here compared to identical samples measured in our neutron study (Shepherd et al., 2022). The values of $k_{\text{surf}}$ for the urban sample are pleasingly close to the values presented by Shepherd et al. (2022). The fitting of the KM SUB model to the data in Fig.7 is not ideal for the remote samples and the authors have more confidence in

the values in the previous study Shepherd et al. (2022), where more temporal resolution was achieved.

It is possible to calculate the uptake coefficient $\gamma$ from the kinetic model output (Shiraiwa et al., 2010) .

$$\gamma = \frac{J_{\text{Ads}} - J_{\text{Des}}}{J_{\text{Coll}}} \tag{9}$$

$\gamma$ is a function of the rate of adsorption, $J_{\text{Ads}}$, rate of desorption, $J_{\text{Des}}$, and rate of OH radical collision with the surface, $J_{\text{Coll}}$,

($J_{\text{Ads}}$, $J_{\text{Des}}$ and $J_{\text{Coll}}$ are defined in Shiraiwa et al. (2010)). The uptake coefficient indicates the rate of OH radical replacement at the surface. As $J_{\text{Ads}}$, and $J_{\text{Des}}$ are dependent on the surface concentration of OH radicals and $J_{\text{Coll}}$ is constant, $\gamma$ indirectly indicates the probability an OH radical will react at a surface. Generally, the value of $\gamma$ is initially larger for the samples collected at remote locations whereas urban samples have smaller initial values of $\gamma$ (Fig. 7). The trend is also reflected in the value of $k_{\text{surf}}$ trend presented in Table 2 and justifies the assumption that $\gamma$ is effectively a measure of the reaction probability

under these conditions.

### 3.4   Material remaining at the air-water interface

There is evidence for a residual film after oxidation with OH radicals (Fig. 7). All kinetic decays appear to level-off after a certain degree of OH radical exposure. The kinetic model fitted to these decay data was adapted to incorporate such a residual film. It is unclear whether the residual film is an unreactive portion of the original film or an oxidised unreactive product (or

even a mixture of both). What is certain is that oxidation has changed film composition, which itself can affect the lifetime of



atmospheric aerosol (Gilman et al., 2004).

### 3.5 Atmospheric implications: Chemical lifetime of organic extracts

It is useful to compare the chemical lifetime of the extracted material with respect to the OH radical oxidation to typical
lifetimes of atmospheric aerosol. Here, the chemical lifetime of the film is defined as the chemical half-life extracted from
the optimised kinetic models for each film at an atmospherically relevant [OH] range (Fig. 8). If the chemical lifetime of
the film is short compared to aerosol atmospheric lifetime, oxidation would need to be considered in atmospheric models.
The lifetime of an atmospheric aerosol can be governed by the relative humidity (Li et al., 2020; Li and Knopf, 2021) of the
environment, the presence (Gill et al., 1983) and composition (Gilman et al., 2004) of a film and by the oxidants present in
the surrounding environment (Knopf et al., 2011). Williams et al. (2002) calculated a lifetime ranging from 1–10 days for an
aerosol with a diameter of $0.1$ $\mu$m at an altitude ranging from 0-8 km. George et al. (2008) sampled ambient aerosol from
Toronto (Canada) and oxidized it in the lab with OH radicals in a flow tube and found OH can oxidise this matter on the
timescales of four days. Likewise Robinson et al. (2006) predicted the timescale for OH oxidation of organic aerosol to be 1-9
days. The chemical lifetimes reported here are very short but in contrast to the studies of George et al. (2008) and Robinson
et al. (2006) the reaction is a surface reaction only. There is no doubt the reaction is fast and should be included in models. The
lifetime of the atmospheric aerosol extract in the study presented was calculated to lie between 48–110 min at $[OH]_{\text{atm}}$ (Fig. 8).
The modelled lifetime is very short in comparison to typical aerosol lifetime of 1–10 days (Kanakidou et al., 2005). Hence, it
can be concluded that atmospheric aerosol films composed of similar material may react very quickly in the atmosphere and
consequently need to be considered in atmospheric aerosol models.

### 415   3.6 Atmospheric implications: Change in top of the atmosphere albedo owing to oxidation of thin film

The presence of a thin organic film can alter the optical properties of an atmospheric core-shell aerosol and this effect is
determined in part by the real and complex refractive indexes and sizes of the core and film (Barker et al., 2023; Donaldson and
Vaida, 2006; Forrister et al., 2015; Gill et al., 1983; Jones et al., 2017; Lack and Cappa, 2010; McGrory et al., 2022; Moffet
and Prather, 2009; Schnaiter et al., 2005; Shepherd et al., 2018). The accumulation of an organic film will alter the ratio of
light that is scattered/ absorbed by a particulate as well as the direction in which this light is scattered (Bond et al., 2013;
Jacobson et al., 2000; Lack and Cappa, 2010; Moffet and Prather, 2009; Shiraiwa et al., 2010; Wu et al., 2014). Furthermore,
owing to substantial differences between the refractive indexes and size distribution of aerosol between environments, the
optical contributions of films are environment-dependent (Barker et al., 2023; Kirchstetter et al., 2004; Kirpes et al., 2019;
Shepherd et al., 2018; Virkkula et al., 2022). As changes are made to organic films, i.e film thickness reduction through
oxidation, the optical properties of the aerosol will also vary; thus, the nature and intensity of these variations are likely also
to be environment-dependent. A combination of core-shell Mie and 1-d radiative transfer modelling has previously been used
to estimate the change in top of atmosphere albedo from uncoated aerosol to coated aerosol of varying film thicknesses and
environments (Barker et al., 2023). Barker et al. (2023) showed that while there were significant differences in how albedo





varied with increasing film thickness between environments, the presence of films on mineral aerosol in both urban and remote
environments caused a decrease in albedo regardless of film thickness for films between 0.1–100 nm. The oxidation-induced
changes to the optical properties of aerosol, and the effect this would have on top of atmosphere albedo, is estimated here
by a combination of Mie and 1-D radiative transfer modelling. The aim was to quantify the albedo impact of changes to film
thicknesses observed in this work, *i.e.*, the oxidation of an organic film is simplified to a halving of film thickness. Particular
attention was made towards films of thickness of 0.1–10 nm owing to the atmospheric relevance of this film thickness range.
Calculations were made using both aqueous and mineral cores and using optical properties of organics and size distributions of
aerosol in urban and remote environments. Variation in the source environment of aerosol resulted in substantial differences in
$\Delta$Albedo between urban and remote aerosol, with effects ranging between +0.001 to +0.014 for urban aerosol and $-0.004 \pm$
0.003 for remote aerosol, with less dependence on the film thickness for remote aerosol. Changes to $\Delta$Albedo are driven by
an interplay of scattering and absorption effects, which were made evident in the variation of the single scattering albedo
and asymmetrical light scattering as initial film thickness was increased. The oxidation of urban aerosol films with increasing
thicknesses resulted in an increase in the single scattering albedo of 0.004 to 0.067 and a decrease of the asymmetry parameter
between 0.000 to 0.022, whereas the oxidation of remote films caused variation in the single scattering albedo of +0.09 to -
0.003 and variation of asymmetry parameter of +0.004 to -0.010. While the core size, core refractive index, initial film thickness
and film refractive index all had some impact on $\Delta$Albedo, the environment-dependent differences in $\Delta$Albedo versus initial
film thickness are driven predominantly by two factors; the imaginary refractive index of material from urban environments is
significantly greater than that from remote environments, and that the film thicknesses are of greater size relative to the core
size of urban aerosol than remote aerosol. There is a distinct difference in changes to the asymmetry parameter and the single
scattering albedo between environments and the lack of significant differences when comparing differences between aerosol
with aqueous and mineral cores. Owing to the optical and physical differences in these aerosols it appears that, typically, the
oxidation of a highly absorbing film on a smaller core (urban) has a greater effect on top of atmosphere albedo than the oxidation
of a less absorbing film of the same thickness on a larger core (remote). However, the effects of oxidation of films < 1 nm in
thickness on $\Delta$Albedo are similar in intensity between urban and remote environments. Barker et al. (2023) demonstrated that
adding a film to mineral aerosol resulted in a decreasing top of atmosphere albedo, hence the removal of the film would cause
an increase in albedo. The effect is also demonstrated for the removal of half the film from mineral as well as aqueous aerosol,
however, the intensity of the effect is much smaller and may be negligible for remote aerosol and an important, but small, effect
for urban aerosol.

## 4 Conclusions

The study presented here demonstrates the successful formation of a thin film of atmospheric aerosol at the air-water interface
from material extracted from atmospheric aerosol. The film was determined to comprise of a two-layer system, indicating
aerosol extracts may produce a film thickness of $\sim 10$ Å for urban extracts and $\sim 17$ Å for remote extracts. In the absence of
other data sources, the film thickness determined in the study could be used in atmospheric radiative transfer models for core-

shell systems. The two-layer system indicates that the atmospheric aerosol extracts may contain hydrophobic and hydrophilic regions. The atmospheric aerosol extracts were exposed to gas-phase OH radicals at a concentration of $3.3 \times 10^6$ molecule cm$^{-3}$ and showed a the bimolecular rate constants of the magnitude of $1 \times 10^{-10}$ cm$^3$ molecule$^{-1}$ s$^{-1}$ and lifetime of the atmospheric
aerosol extract at the air-water interface to be calculated as short at $\sim 1$ hour Optimised KM sub kinetic models suggested that the atmospheric lifetime of the reactive component of the films studied can vary between minutes and days depending on the ambient hydroxyl radical concentration. The determined film thickness and film lifetimes presented in this work represent values for a large ensemble of molecules and give an estimate of the values. However, these values do not represent an average, the average macroscopic scale film properties at the air-water interface of a trough may not be statistically applicable to
macroscopic areas of individual aerosol droplets. The removal of half of the organic film from the air-water interface may have an increasing effect on the top of the atmosphere albedo for urban films and a neglegible effect for remote films.

*Code availability.*  The code for the kinetics modelling is available doi: 10528/zenodo.11962921 as a series of Jupyter notebooks The MultilayerPy software used in this work, including tutorials and documentation, is available at https://github.com/tintin554/multilayerpy and https://doi.org/10.5281/zenodo.6411188 (Milsom et al., 2022a). The code is released under the GPL v3.0 license..

*Data availability.*  Data is available at doi: 10528/zenodo.11962921

*Sample availability.*  All samples were consumed in the experiment.

*Author contributions.*  Rosalie H. Shepherd conducted all experiments, extracted all atmospheric aerosol extracts, analysed and interpreted data collected and wrote the paper. Martin D. King, Andrew D. Ward conceived the experiment idea and assisted during the experiment. Neil Brough collected samples from Antarctica. Thomas Arnold was the synchrotron beamline scientist who assisted with the experiment
and reduced all data. Adam Milsom performed the surface kinetic modelling, which was analysed and interpreted with input from Christian Pfrang. Edward Stuckey, Martin King and Rebecca Welbourn conceived the modelling and it was performed by Edward Stuckey

*Competing interests.*  The authors declare that they have no conflict of interest

*Acknowledgements.*  The authors would like to thank Diamond Light Source for granting access time on I07 under the experiment number SI13493 and NERC NE/T00732X/1. Rosalie H. Shepherd would like to thank STFC for funding the student grant ST\L504279\1. The authors
would also like to thank Jerry Morris (RHUL) for manufacture of the Langmuir trough.





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





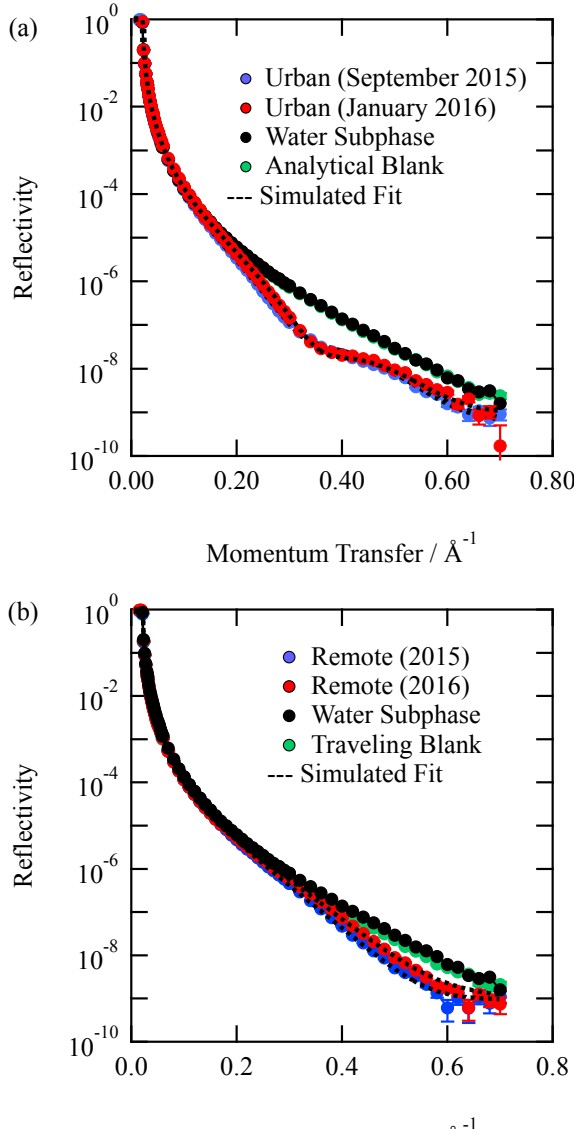

**Figure 1.** Typical x-ray reflection profiles for (a) urban atmospheric aerosol extract and (b) remote atmospheric aerosol extract. A corresponding analytical (or travelling blank) is displayed in each figure. In both panels the analytical blank is indistinguishable from the water sub-phase with a clean air-water interface within error The error bars represent statistical counting errors from accumulation and binning of photons, and are generally smaller the points depicted.





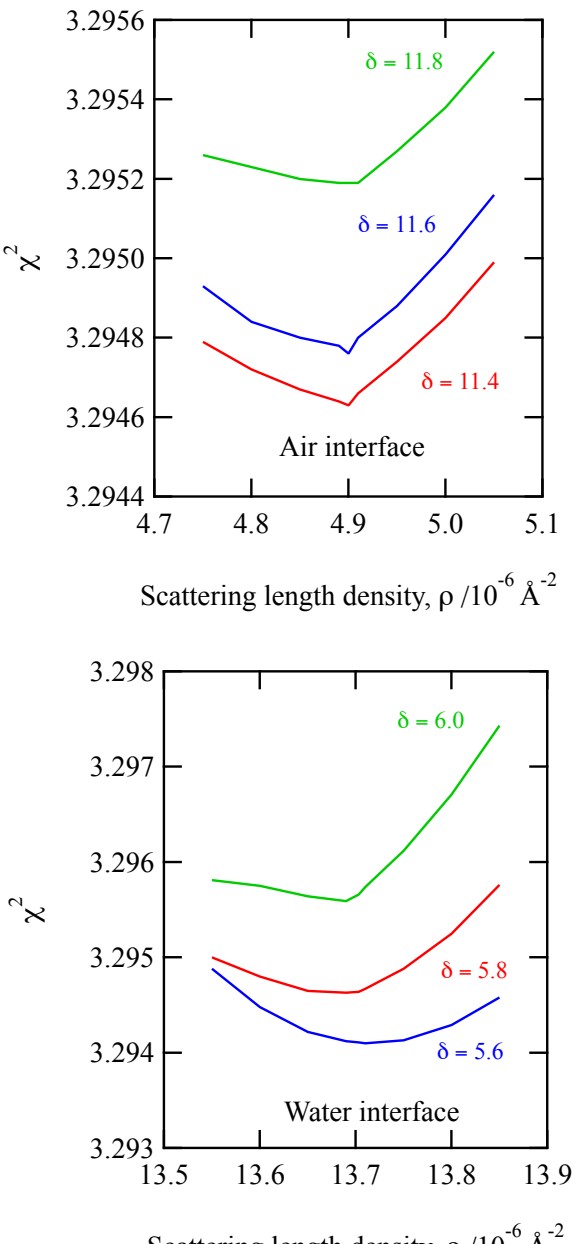

**Figure 2.** A figure of merit for reproducing the x-ray reflectivity profile by a computed profile relative to the experimental data as a function of the value of the computed scattering length density for different film thicknesses. Note the minimum of the figure of merit for different scattering length densities seem insensitive to the thickness in the range of values presented. Both panels are for Urban September 2015 sample. The top panel corresponds to the air interface and the bottom panel corresponds to water interface.





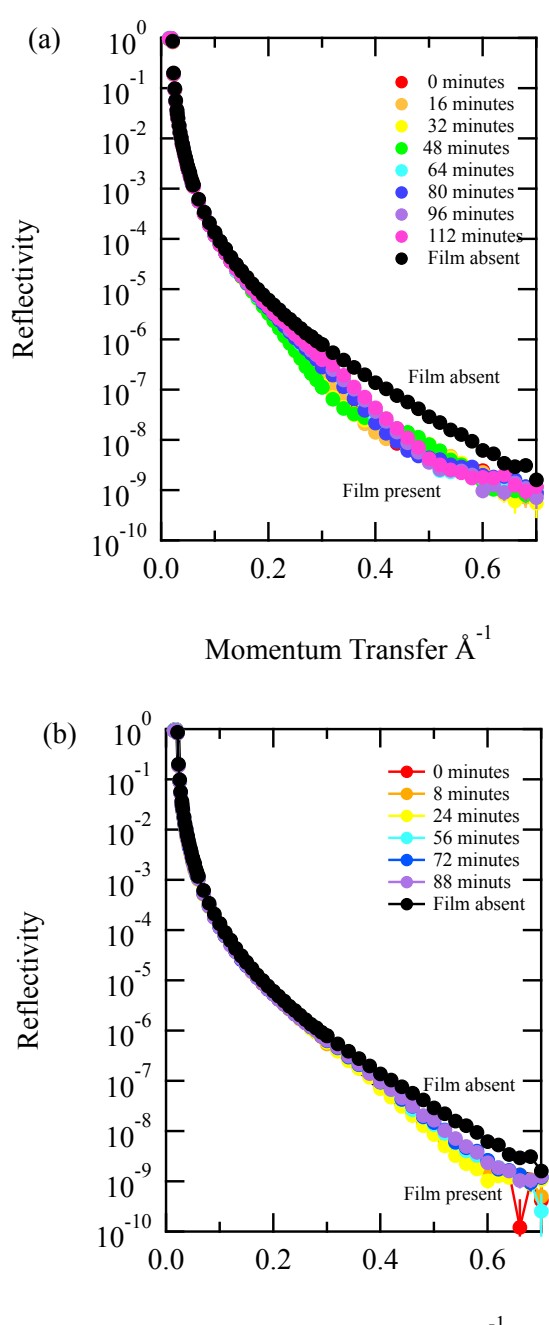

**Figure 3.** X-ray reflection profiles of a film of urban atmospheric aerosol (collected September 2015) as the film was exposed to gas-phase OH radicals. Upon exposure to gas-phase OH radicals, the reflectivity of the film increases indicating a reaction is occurring at the air-water interface.



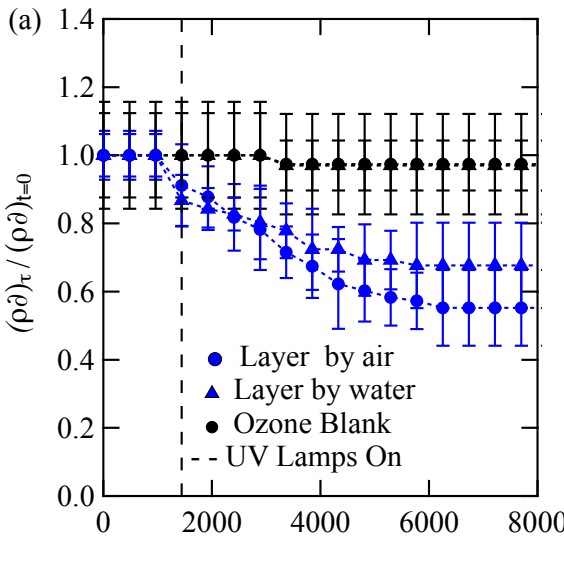

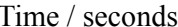

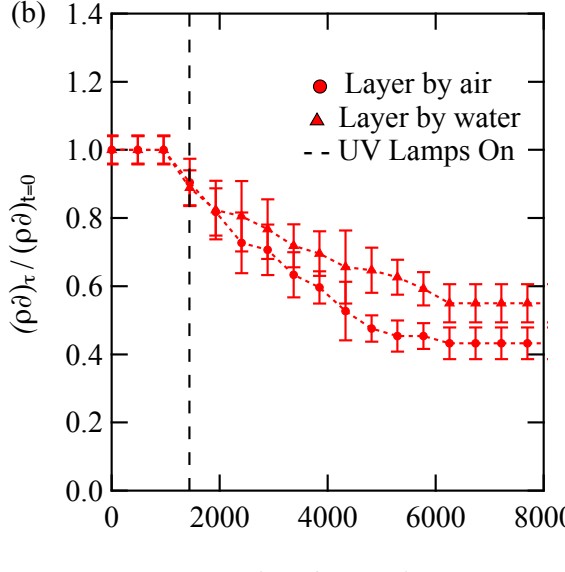

**Figure 4.** The kinetic decay of the scattering length per unit area for an organic film or Urban atmospheric aerosol extract collected during (a) September 2015 and (b) January 2016 decayed when exposed to hydroxyl radical. $[OH] = 3.3 \times 10^6$ molecule cm$^{-3}$



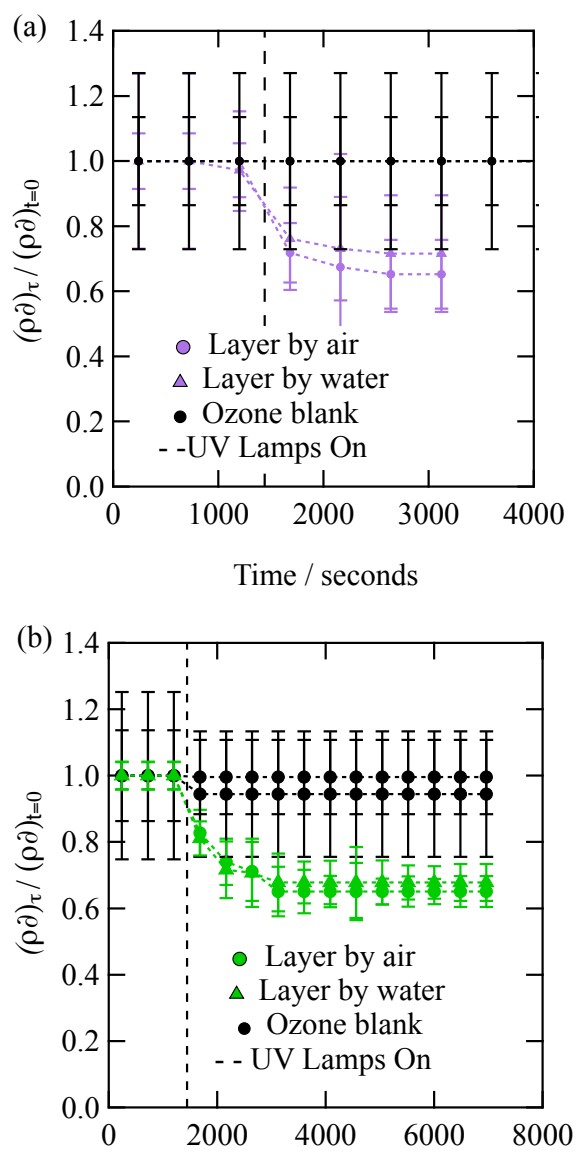

**Figure 5.** The kinetic decay of the scattering length per unit area for an organic film of Antarctic atmospheric aerosol extract collected during (a) 2015 Antarctic summer and (b) 2016 Antarctic summer decayed when exposed to hydroxyl radical. $[OH] = 3.3 \times 10^6$ molecule cm$^{-3}$. UV lamps switched on at 24 minutes.





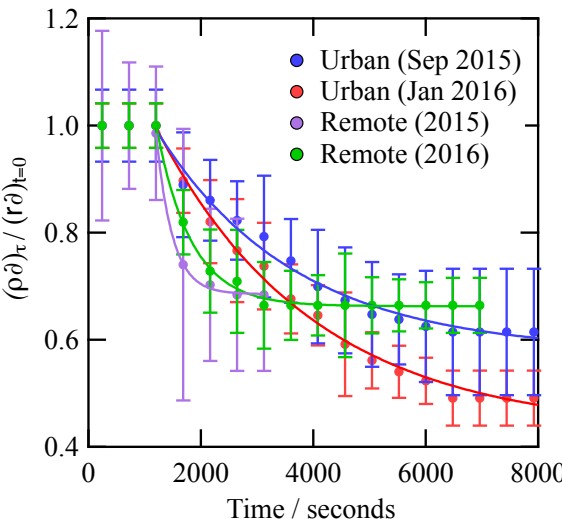

**Figure 6.** The kinetic decay of the scattering length per unit area for an organic film or Urban atmospheric aerosol extract collected during (a) September 2015 and (b) January 2016 decayed when exposed to hydroxyl radical. $[OH] = 3.3 \times 10^6$ molecule cm$^{-3}$.





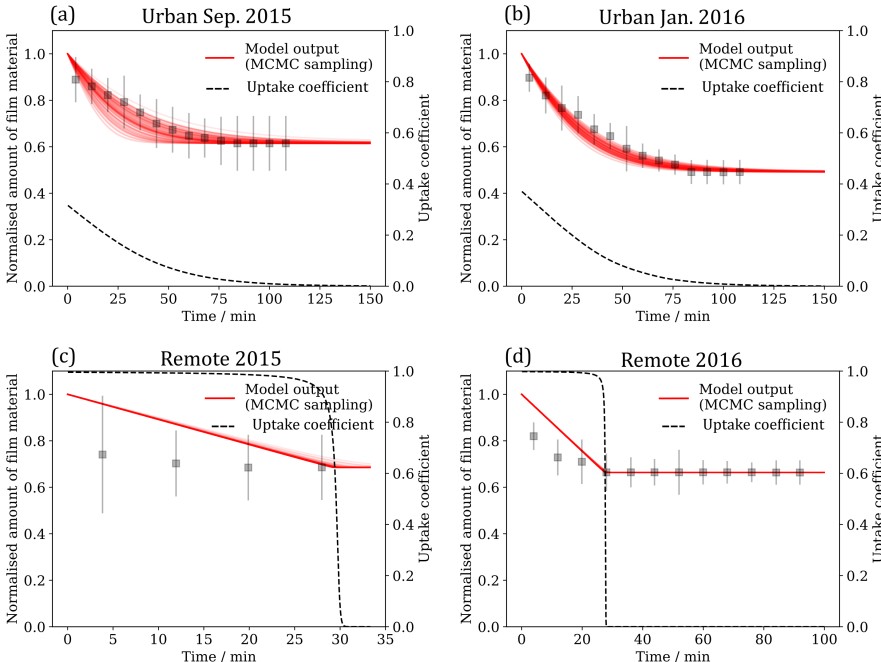

**Figure 7.** Results from the fitting the kinetic model (red) described in section 3.3 to the experimental data (grey). The range of model outputs from the MCMC sampling procedure consistent with the data are presented. The evolution of the uptake coefficient over time is extracted from the best fitting model run for each sample.



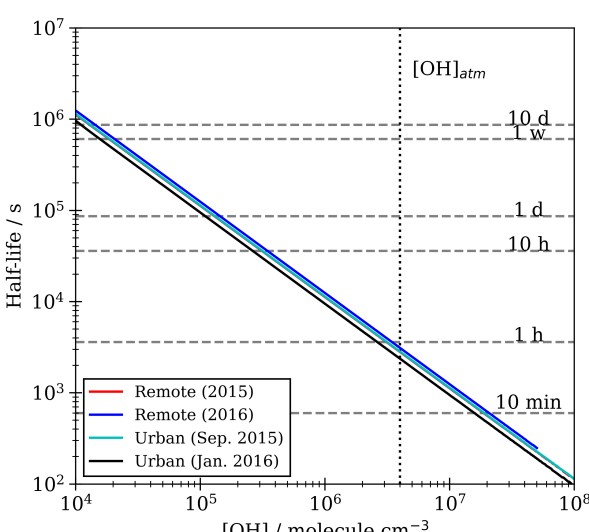

**Figure 8.** The half life of the film owing to chemicals oxidation by OH radical for different OH radical mixing ratios. A typical hydroxyl radical mixing ratio [OH]$_{atm}$ is shown as a vertical dotted line.





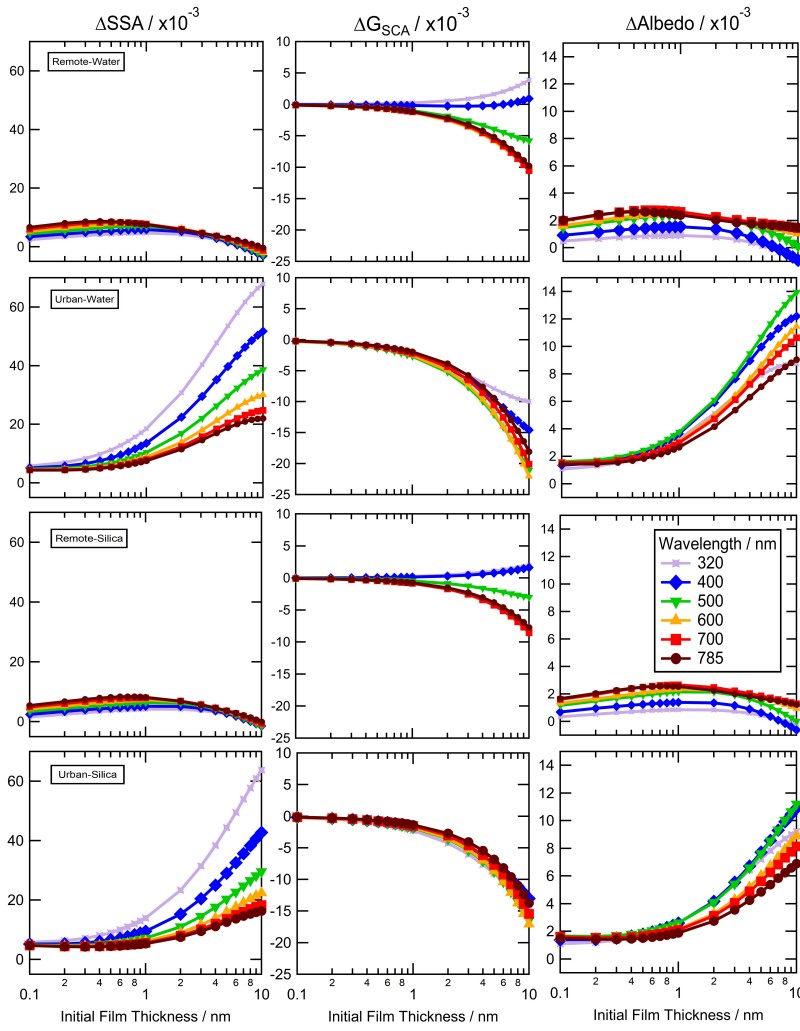

**Figure 9.** Change in aerosol single scattering albedo, ΔSSA, asymmetry parameter, $\Delta g_{sca}$, and top of atmosphere albedo, ΔAlbedo, (80 km altitude, 60° Solar zenith angle) calculated using a combination of Mie and 1-D radiative transfer modelling for the impact of the oxidation (represented by a halving of film thickness) of organic coated spherical silica and aqueous aerosol from urban and remote environments for wavelengths of light between 320–785nm. Core and film materials were given complex refractive indexes representative of the material/environment. The single scattering albedo and asymmetrical light scattering of these four types of aerosol were calculated for aerosol with core sizes between 10–1000 nm with initial film thicknesses 0.1–10 nm, and normalised by the size distribution of aerosol from their respective environments. The single scattering albedo and asymmetry parameter were calculated for aerosol before and after oxidation. Radiative transfer models were produced for each aerosol disturbution, for each film thickness before and after oxidation. Values of single scattering albedo and asymmetry parameter were applied to a 3 km thick layer of aerosol at the planetary boundary layer, and top of atmosphere albedo was determined for a given model using aerosol with a given film thickness. ΔSSA, $\Delta g_{sca}$ and ΔAlbedo are the difference between the post-oxidation models from the pre-oxidation models.



**Table 1.** Table of the x-ray scattering length density ($\rho$) and film thickness ($\delta$) for urban and remote atmospheric aerosol extract thin films at the air-water interface.

|  | Sample | Layer | $\rho$ | $\delta$ |
|---|---|---|---|---|
|  | . |  | $/10^{-6}$ Å$^{-2}$ | / Å |
| Urban | September 2015 | Air Interface | $4.88 \pm 0.06$ | $11.6 \pm 0.2$ |
|  |  | Water interface | $13.70 \pm 0.01$ | $5.9 \pm 0.2$ |
|  |  | Total |  | $17.5 \pm 0.3$ |
|  | January 2016 | Air Interface | $5.33 \pm 0.9$ | $9.8 \pm 0.6$ |
|  |  | Water interface | $13.96 \pm 0.18$ | $6.6 \pm 0.4$ |
|  |  | Total |  | $16.4 \pm 07$ |
| remote | 2015 | Air Interface | $2.55 \pm 0.05$ | $4.5 \pm 0.3$ |
|  |  | Water interface | $10.91 \pm 0.17$ | $6.2 \pm 0.1$ |
|  |  | Total |  | $10.7 \pm 0.3$ |
|  | 2016 | Air Interface | $2.84 \pm 0.08$ | $5.4 \pm 0.6$ |
|  |  | Water interface | $10.96 \pm 0.01$ | $4.3 \pm 0.1$ |
|  |  | Total |  | $9.7 \pm 0.6$ |




**Table 2.** Pseudo first order and bimolecular rate constants as well as atmospheric film lifetime for the reaction between atmospheric aerosol extract and gas-phase OH radicals. To determine the lifetime of the atmospheric aerosol extracts when in the atmosphere, the concentration of $[OH]_{atm}$ was determined from literature to be $4 \times 10^6$ cm$^{-3}$ (Frost et al., 1999; George et al., 1999).

|  | Sample | $k'$ / $10^{-4}$ $s^{-1}$ | [OH] / $10^6$ molecule cm$^{-3}$ | $k_4$ / $10^{-10}$ cm$^3$molecule$^{-1}s^{-1}$ | $\tau$ / minutes | $k_{surf}$ / $10^{-8}$cm$^2s^{-1}$ |
|---|---|---|---|---|---|---|
| Urban | September 2015 | $4.04 \pm 0.33$ | 3.3 | 1.2 | 34 | $21.3 \pm 8.9$ |
|  | January 2016 | $4.04 \pm 0.33$ | 3.3 | 1.1 | 37 | $16.1 \pm 2.2$ |
| remote | 2015 | $34.6 \pm 2.5$ | 3.3 | 10.5 | 4 | $4950 \pm 2820$ |
|  | 2016 | $16.2 \pm 0.6$ | 3.3 | 4.9 | 9 | $5700 \pm 2490$ |



**Table 3.** A comparison of the properties of similar films analysed in the work presented here and taken from Shepherd et al. (2022). Film thickness, bimolecular reaction coefficient for reaction (1) and $k_{\mathrm{surf}}$ from the KM SUB kinetic modelling. These film are comparable but not identical. The values of the quantities from the x-ray reflection work (presented here) are marked with XR and the values from the Neutron reflection work (NR) and from Shepherd et al. (2022).

| Property | Remote 2015 | Remote 2016 | Urban (Sep 2015) | Urban (May 2015) | Urban (Jan 2016) |
|---|---|---|---|---|---|
| Film thickness / Å (NR) | 7.6 | 10.5 | | 6.63 | 10.2 |
| Film thickness / Å (XR) | 10.7 | 9.7 | 17.5 | | 16.4 |
| $k_1/10^{-10}\mathrm{cm^3 molecule^{-1}s^{-1}}$ (NR) | 1.4 | 0.93 | | 1.3 | 1.5 |
| $k_1/10^{-10}\mathrm{cm^3 molecule^{-1}s^{-1}}$ (XR) | 10.5 | 4.9 | 1.2 | | 1.1 |
| $k_{\mathrm{surf}}/10^{-8}\mathrm{cm^2 s^{-1}}$ (NR) | 50 | 9.3 | | 5000 | 23 |
| $k_{\mathrm{surf}}/10^{-8}\mathrm{cm^2 s^{-1}}$ (XR) | 4950 | 5700 | 21 | | 16 |