# Peer review of "The lifetimes and potential change in planetary albedo owing to the oxidation of thin surfactant organic films extracted from atmospheric aerosol by hydroxyl (OH) radical at the air-water interface of particles"

_EGUsphere, 2024_

## Author Comment (AC1)

**The lifetimes and potential change in planetary albedo owing to the oxidation of organic films extracted from atmospheric aerosol by hydroyxl (OH) radical oxidation at the air-water interface of aerosol particles**

Rosalie Shepherd, Martin King \*, Andrew Ward, Edward Stuckey, Rebecca Welbourn, Neil Brough, Adam Milsom, Christian Pfrang, and Thomas Arnold

We thank both reviewers for their time and comments, which have helped improve this paper.

**RC1: 'Comment on egusphere-2024-2367', Anonymous Referee #1, 15 Sep 2024**

This manuscript reports X-ray reflectivity (XR) measurements of organic material extracted from authentic aerosol samples and applied to a liquid water surface. XR yields information on the thickness, the electron density and the vertical structure of the film. The aerosol samples were collected in an urban environment to represent polluted air and in a remote location in Antarctica to represent clean air. The XR measurements indicated the resulting films on water to be less than 2 nm thick and application of fits to simulate the XR profiles suggest a 2-layer structure, the one on the water-side of the film exhibiting higher electron density than the one on the air-side, indicating hygrophilic functionalities towards the water surface. The latter was more pronounced for the sample from the remote location. Exposure of the films to OH lead to degradation of the film at a rate close to the OH collision limit and a change in the structure. This was supported by multilayer modelling to obtain an apparent surface reaction rate coefficient for OH with the film. In addition, model calculations are presented to demonstrate that expected changes to film thickness and optical properties due to aging (thus degradation of organic films) will affect aerosol scattering albedo globally.

The work is clearly interesting and provides insight into the surface properties of organic material likely to assemble on atmospheric aerosols. The methods used are unique and the direct conclusions from the XR measurements appear sound. It is clear that, given limited access to synchrotron X-ray facilities, not a larger amount of samples could be investigated to expand the scope of the work. More specific questions and concerns related to these are given below. On a more general level, the calculations with respect to the optical properties appear rather disconnected from the other parts of the work. Film thicknesses measured on water in this work cannot be used to infer film thicknesses in atmospheric particles (see details below), nor can their life time from the measurement on the water surface be extrapolated directly. Therefore, the claim of the impact of aging of surface layers on Earth's albedo are rather speculative, and this section of the manuscript is not really useful for assessing the implications of the present work. Those calculations would likely be better presented in a separate manuscript, where changes to optical properties of particles due to aging are reviewed based on a broad literature overview, then used to assess potential impacts on aerosol optical properties.

We would respectfully challenge the reviewer's assertion that these film thickness' cannot be applied to the atmosphere . We suggest that the difference in thinking may come down to a consideration of 'thick' or 'thin' films. Here we are clearly in the thin film domain, so much so that we have added the word "thin" to the paper title. In essence we believe we are describing the chemical oxidation of thin surfactant films at the air-water interface. These are not thick filmed coreshell particles that have phase separated. We provide more detail to the reviewer below and demonstrate that our work when coupled with modelling may be used to estimate the lifetime of thin organic films on aqueous aerosol and hydrometeors. The radiative-transfer calculations which demonstrate the oxidation of such thin films may not be negligible. We believe that this is an important part of our paper and outlines the potential atmospheric effects from their oxidation. The authors of the current work feel it is important to use simple models to apply results from the laboratory to the atmosphere so that more advanced studies can be contemplated and considered by others.

**Specific comments**

Somewhere in the experimental information sections, the authors should give more quantitative information on how much organic was extracted and deposited onto the water in the trough. Over what area was the material spreading? Do the authors assume that it fills the entire water surface area, or is it forming a floating island? Was the mass deposited similar for the two samples? or what is the basis for the comparison of thickness derived later from XR if it cannot be normalized to mass? Have the authors done some calibration experiments with well defined surfactants with different properties, such as fatty acids or similar? Specifically, on line 134, what was the x-ray beam footprint?

The following pieces of text have been added to the manuscript to clarify that we are studying thin films at the air-water interface formed from surfactants and not thick films that may be formed from phase separation. We also include an estimate of the surface coverage (molecule m-2), and mass of the films added.

The introduction: "The thin films described in the work here are typically monolayers or a few molecules thick and are formed by molecules with surfactant behaviour (Davies and Rideal 1961). i.e. favouring the air-water interface. The thicker films formed by phase separation are not considered."

In section 2.2 Experimental set up - "These solutions were spread onto water using a microliter syringe to produce an approximate surface pressure of  $\sim 15$  mN m-1 as determined in off-line experiments with the same trough and a surface tensiometer. An excellent and detailed experimental and theoretical consideration of spreading insoluble surfactants at an air-water interface can be found in Davies and Rideal (1961). A similar surface pressure produced by either a monolayer of fatty acid molecules like oleic acid would give a surface coverage of  $2 \times 10^{18}$  molecule cm-2, King et al (2010), about 16 µg of material added to the trough described here, or by using the mass density of spreading solution ( $^{\sim}7 \times 10^{-10}$ 5 g ml-1 cm-3) of similar aerosol extract Shepherd et al.(2018) to those used here demonstrates about 10µg of materials was added to the trough. The material covers the entire surface of the trough,  $\sim 175 \text{ cm}^2$ , adding more of the material does not produce a large increase in the thickness of these films as shown by figure 4 of Shepherd et al., (2022) which shows the film thickness of a related urban sample starts to asymptote as more material was added. Adding too much material to the interface produced a thin film with visible lenses of material. Similar behaviour is shown by piston oils such as oleic acid (Gaines, 1966)."

J. T. Davies and E. K. Rideal, Interfacial Phenomena, Academic Press, 1961

**G.L. Gaines, Insoluble monolayers at liquid gas interfaces, Interscience Publishers, 1966**

In section 3.1 x-ray reflection and properties of the film "It should be noted that the work presented here confirms that these extracted samples form stable thin films at the air-water interface those thickness does not increase notably with the addition of more material, but results in the formation of lenses (Gaines 1966) like the related films in other related studies (Shepherd 2020 and Stuckey 2024). However, it is useful to contrast the films presented here with the woodsmoke sample of Stuckey et al. (2024) which is unusual in producing much thicker films stabilised by differentiating the thick film into three separate layers. The work of Stuckey et al (2024) demonstrates that the reflection techniques would be sensitive to thick film formation if it were occurring in the work presented here."

XRR, as a technique, is well established as a method that averages microscopic structure perpendicular to the surface over macroscopic length scales in the plane of the surface. The x-ray beam footprint is a function of the beam size (<200  $\mu$ m) and Q as the angle of x-ray incidence ( $\theta$  in equation I) is changed during acquisition of x-ray profiles like those shown in Figure 1. At very small angles, low-Q below the critical angle, the x-ray footprint is a bit longer than the trough, but it quickly reduces to be comparable or smaller than the size of the trough (20cm). Over the full reflectivity curve up to 0.6A-1, the beam goes from about 230mm to <5mm projected onto the surface. This requires a geometrical footprint correction to be applied for the first few points during data reduction to correct the reflectivity for over-illumination. The beam has an approximate gaussian profile which is projected onto the surface. The film is uniform laterally across the trough as evidenced by the control experiments (without ozone or without UV light). The x-ray footprint is long and narrow and moved horizontally (perpendicular to the beam scattering plane) across the trough for the acquisition of each x-ray reflectivity profile to avoid beam damage. In the control experiments these profiles are identical demonstrating that the surface is uniform to x-ray reflectivity. The following text has been added

"X-ray reflection as a technique, is well established as a method that averages microscopic structure perpendicular to the surface over macroscopic length scales in the plane of the surface. The x-ray footprint was comparable to the length of the trough (200mm) at low angles and approx.  $200\mu$ m wide. It was moved horizontally across the trough (perpendicular to the beam scattering plane) between measurements of the x-ray reflectivity profile to prevent beam damage. Further information on the beam footprint is clearly described in Salah et al. (2007)."

Salah, F., Harzallah, B. & van der Lee, A. (2007). J. Appl. Cryst. 40, 813-819.

And

"As evidenced by the control experiments, labelled 'ozone blank' on fig.s 4 and 5, the x-ray reflectivity profile did not change demonstrating little or no reaction with ozone and that the film on the trough is horizontally homogenous. The trough is moved between the measurements of x-ray reflectivity profile to prevent beam damage."

line 205, equation (5): this rate law is a bit strange, as [OH] is provided as gas phase concentration. What is the meaning of the value for k4? OH and the molecules (presented as surface coverage) are not homogeneously mixed. This equation should either contain the collision rate, or the OH concentration treated as a surface concentration, as the authors do when they use the mulilayer model.

We thank the reviewer for this question which has motivated the authors to consider with care whether the bimolecular kinetics are necessary, as the kinetics described by the KM-SUB, or multilayer model are superior. We have chosen to remove the bimolecular kinetics from the paper.

line 243: here comes the point of transferring the thickness retrieved from the water surface by XR to that of atmospheric aerosols. First, the extraction only takes a fraction of surface active material from the sample. 2nd, there is no calibration of mass of organic vs thickness in the trough: on the water surface, the extract spreads until it obtains an equilibrium configuration on an extended water surface. In the atmosphere, the same material may by forced into a separate phase forming a much thicker layer.

The following pieces of text have been added to the manuscript to clarify that we are studying thin films at the air-water interface formed from surfactants and not thick films that may be formed from phase separation. We also include an estimate of the surface coverage (molecule m-2), and mass of the films added.

The introduction: "The thin films described in the work here are typically monolayers or a few molecules thick and are formed by molecules with surfactant behaviour (Davies and Rideal 1961). i.e. favouring the air-water interface. The thicker films formed by phase separation are not considered."

In section 2.2 Experimental set up - "These solutions were spread onto water using a microliter syringe to produce an approximate surface pressure of  $\sim 15$  mN m-1 using off-line experiments with the same trough and a surface tensiometer. An excellent and detailed experimental and theoretical consideration of spreading insoluble surfactants at an air-water interface can be found in Davies and Rideal (1961). A similar surface pressure produced by a monolayer of fatty acid molecules like oleic acid would give a surface coverage of  $\sim 2 \times 10^{18}$  molecule cm-2, King et al (2010) or about 16 µg of material added to the trough described here. Using the mass density of spreading solution (~7×10-5 g ml-1 cm-3) of similar aerosol extract Shepherd et al.(2018) to those used here demonstrates about 10µg of materials was added to the trough. The material covers the entire surface of the trough, ~175 cm2, adding more of the material does not produce a large increase in the thickness of these films as shown by figure 4 of Shepherd et al., (2022) which shows the film thickness of a related urban sample starts to asymptote as more material was added. Adding too much material to the interface produced a thin film with visible lenses of material. Similar behaviour is shown by piston oils such as oleic acid (Gaines, 1966). Using the mass density in Shepherd et al (2018) or a comparable film of oleic acid"

J. T. Davies and E. K. Rideal, Interfacial Phenomena, Academic Press, 1961

G.L. Gaines, Insoluble monolayers at liquid gas interfaces, Interscience Publishers, 1966 line 368, back to kinetics: as mentioned above, the meaning of k4 is not clear. This is not a homogeneous reaction. What is the diffusion limit? for gas phase? do you mean the collision limit?

**See above the bimolecular kinetics are not needed with the multilayer model and have been removed**

line 389: How it is possible to have an apparent loss rate that then translates to an uptake coefficient so close to one? Given the geometry of the setup, this should be gas phase diffusion limited and thus considerably lower.

**After consideration this line has been removed.**

line 414: applying OH uptake observed to degradation of organics in the real atmosphere: in the atmosphere, oxidation targets are not at the surface only, while the reaction with gas phase OH is limited to surface and by the supply of OH from the gas phase, which is in the end determining the lifetime. This comes back to the point made above: if in the real aerosol, the organics are forming a thicker phase, their degradation (on average) is much slower.

**The work presented here is for thin films, not thick films.**

in the last lines of the manuscript, line 468 - 470, the authors reflect on these concerns, but still, the last sentence reiterates the impact on albedo.

The text in the last line has been edited to make it clear it is for thin films.

**Technical comments**

line 46: something is wrong with this sentence: 'oxidation between sub-micron aerosols'?

**Corrected to 'oxidation of sub-micron aerosols'**

lines 74-79: consider splitting this overly long sentence into 2 or 3. From 'to study oxidation...' to 'by aqueous-phase OH' takes too long, and the reader looses orientation, at least this reviewer.

**Corrected: sentence broken into two smaller sentences.**

line 101: the filters were not collected, the samples on the filters were collected.

The text has been changed to "Organic materials from atmospheric aerosols was extracted from quartz fibre filter. The filters were from two locations:"

line 124: ... in the dark until use for the X-ray reflectivity measurements. Not clear at this point what a beam line is.

We use the reviewer's suggested text of ": ... in the dark until use for the X-ray reflectivity measurements".

line 141: already mention here 'Generation of gas-phase OH radical by O3 photolysis...'

The reviewer's suggested text has been added "...by ozone photolysis...".

line 224: a small side remark: if the model indicates photolysis of molecular O2 contributing to OH production, can that be from 254 nm radiation or is there some 185 nm light from the Hg discharge leaking through the lamp tube?

The upper threshold wavelength for molecular oxygen photolysis is around 242 nm (Okabe,H., "Photochemistry of small molecules", Wiley, 1978), and the photolysis of molecular oxygen is, as the reviewer points out, owing to our lamps producing some light at a wavelength of 185nm. The following text has been added:

"The mercury discharge lamps were stated as 'ozone-free', although in practice not all of the radiation of wavelength 185 nm is removed by the lamp tube."

line 232: why are the authors running a global optimisation algorithm if only one parameter was varied?

Corrected. The text now reads "A differential evolution (Storn and Price (19197) was used to optimise the..."

line 290: state more explicitly which type of functional groups exhibit higher electron density and which ones lower.

The following text has been added: "... *e.g.* a phosphocholine lipid molecule like 1,2-Dipalmitoylphosphatidylcholine would have a 'head' group of Phosphatidylcholine ( $(CH_3)_3NCH_2CH_2OP(O_2)O$ -) and a 'tail' of two palmitic acids ( $C_{16}H_{32}O_2$ )

line 470: to microscopic areas, not macroscopic, isn't it.

Corrected. Text now reads: "....may not be statistically applicable to individual aerosol droplets."

Citation: https://doi.org/10.5194/egusphere-2024-2367-RC1

**RC2**: 'Comment on egusphere-2024-2367', Anonymous Referee #2, 09 Oct 2024 In this manuscript, the authors present a beamline study of organic films at the waterair interface, monitoring their X-ray reflectivity as a function of oxidation by gas-phase hydroxyl radical. A strength of the work is that the films were prepared from extracts of ambient aerosols, collected on the urban campus and the remote Antarctic research station. In addition to the direct measurements of evolution, results of kinetics and radiative transfer modelling are presented. An important conclusion is that the oxidation of the thin films is rapid, consistent with previous measurements, which together provide transferrable insights into the evolution of organic aerosol in the atmosphere. The manuscript is suitable for publication in ACP. I have only minor comments for the authors to consider.

Title: I like the comprehensive title that includes lifetimes and planetary albedo, but some terms are redundant. I recommend revising it so that aerosol and oxidation are both mentioned only once.

Corrected: The second 'oxidation' and 'aerosol' are removed,: "The lifetimes and potential change in planetary albedo owing to the oxidation of thin organic films extracted from atmospheric aerosol by hydroxyl (OH) radical at the air-water interface".

28: The introduction of core-shell particles through liquid-liquid phase separation here is helpful context. Several of my comments below are about the need to contrast the film thickness in the trough with the thickness of water-insoluble species in core-shell particles, so that could begin here.

The following pieces of text have been added to the manuscript to clarify that we are studying thin films at the air-water interface formed from surfactants and not thick films that may be formed from phase separation. We also include an estimate of the surface coverage (molecule m-2), and mass of the films added.

The introduction: "The thin films described in the work here are typically monolayers or a few molecules thick and are formed by molecules with surfactant behaviour (Davies and Rideal 1961). i.e. favouring the air-water interface. The thicker films formed by phase separation are not considered."

In section 2.2 Experimental set up – "These solutions were spread onto water using a microliter syringe to produce an approximate surface pressure of ~15 mN m-1 using off-line experiments with the same trough and a surface tensiometer. An excellent and detailed experimental and theoretical consideration of spreading insoluble surfactants at an air–water interface can be found in Davies and Rideal (1961). A similar surface pressure produced by a monolayer of fatty acid molecules like oleic acid would give a surface coverage of  $\sim 2 \times 10^{18}$  molecule cm-2, King et al (2010) or about 16 µg of material added to the trough described here. Using the mass density of spreading solution ( $\sim 7 \times 10^{-5}$  g ml-1 cm-3) of similar aerosol extract Shepherd et al.(2018) to those used here demonstrates about 10µg of materials was added to the trough. The material covers the entire surface of the trough,  $\sim 175$  cm2, adding more of the material does not produce a large increase in the thickness of these films as shown by figure 4 of Shepherd et al., (2022) which shows the film thickness of a related urban sample starts to asymptote as more material was added. Adding too much material to the interface produced a thin film with visible lenses of material. Similar behaviour is shown by piston oils such as oleic acid (Gaines, 1966). Using the mass density in Shepherd et al (2018) or a comparable film of oleic acid"

J. T. Davies and E. K. Rideal, Interfacial Phenomena, Academic Press, 1961

**G.L. Gaines, Insoluble monolayers at liquid gas interfaces, Interscience Publishers, 1966**

73: The importance of studies on ambient aerosol extracts is expressed well here. 83: This statement of purpose should be revised. I mention it below, too, but the emphasis here on film thickness for organic matter at the air-liquid interface makes this goal sound like a broad, transferrable property - but my understanding is that this is dictated by technical experimental constraints, i.e., mass extracted, mass dispensed onto the trough, and the trough dimensions. The third and four purposes are broader and more atmospherically relevant, so those should be the focus here.

We have added text above to highlight these are surfactant thin films and not thick films. These films are broadly independent of trough geometry and dimensions and as now stated above the amount added was to reproduce a surface pressure of ~15 mN m-1.

88: Related to my previous comment, the film thicknesses determined experimentally relate to the sample mass and density and trough geometry and seem to have little bearing on the thickness of water-insoluble, hydrophobic organic components in coreshell particles, e.g., from liquid-liquid phase separation, which will depend on the mass distribution of organic species across the polarity scale. The observed kinetics is what is more transferrable to the planetary albedo discussion.

We have added the following text to the introduction: "The thin films described in the work presented here are typically monolayers or a few molecules thick and are formed by molecules with surfactant behaviour (Davies and Rideal 1961). i.e. favouring the air-water interface. The thicker films formed by phase separation are not considered."

142: Here it may be worthwhile stating explicitly that the RH is 100%.

Corrected: The following text has been added: " ...included in the Tedlar bag and the relative humidity was taken as 100%."

222: In the future, could UV illumination-only controls be performed under nitrogen?

We will do this for our next photolysis experiment.

255: This range is reasonable, but it may be helpful placing it in the context of the small film thicknesses determined experimentally, on the order of 1 nm, i.e., on the low end of this range.

The following text has been added: "...noting that the experimental measurement of thickness presented here are or order 1nm, i.e. towards the smaller end of the range studied."

313: This sentence relates to my previous comments on film thickness.

See above - no revision needed.

319: Components here suggests distinct molecules in a mixture, so it could be replaced with regions or functional groups, etc., to be consistent with amphiphilic components.

The following text has been added: "... *e.g.* a phosphocholine lipid molecule like 1,2-Dipalmitoylphosphatidylcholine would have a 'head' group of Phosphatidylcholine ( $(CH_3)_3NCH_2CH_2OP(O_2)O$ -) and a 'tail' of two palmitic acids ( $C_{16}H_{32}O_2$ )"

330: This statement should be revised to contrast film thicknesses in the trough with those in ambient particles.

See above - no revision needed.

352: The curves in Figure 4 do seem to suggest a weak dependence on the layer. In both panels, the decay for the layer by air is slightly more significant that that for the layer by water. In panel b, the curves are separated, even accounting for the experimental variance in the error bars. Could this observation be connected to the accessibility of the layers, e.g., potentially large hydrophobic groups obstructing access of OH to the lower hydrophilic groups?

Whilst it may be very tempting to draw this conclusion, the error bars on the figure 4 do not allow this interpretation. However, the weak dependence is thought provoking and this this is a topic of investigation for us.

361: Perhaps first mention the recalcitrant fraction here.

The bimolecular analysis has been removed from the paper as it is superseded by the more advanced multi-layer KM-SUB model, and this whole paragraph has been removed along with the need to mention the recalcitrant fraction here.

364: Throughout,  $k_1$  and  $k_4$  are used interchangeably - e.g.,  $k_4$  is used here and in Table 2, and  $k_1$  is used in Table 3 - so one should be corrected.

Corrected. Reaction numbers have been checked throughout the manuscript.

369: How relevant is this diffusion limit here? I usually think of this for bulk solution reactions rather than multiphase reactions at an interface.

The bimolecular analysis has been removed from the paper as it is superseded by the more advanced multi-layer KM-SUB model, and consideration of such a diffusion limit is irrelevant.

376: The language here is unnecessarily vague. A more specific comparison would be helpful and justify the sentence on line 379.

Corrected: We have provided numerical comparison in Table 3 between the neutron and x-ray studies, but had not linked to it from the text. The text now reads "Fitted values of  $k_{surf}$  for reaction of the OH radical with the remote samples are generally larger in the study presented here compared to identical samples measured in our neutron study (Shepherd et al., 2022) and are compared in table 3. Table 3 contains values of  $k_{surf}$  for reaction of OH radical with thin films derived from atmospheric aerosol determined from this study and a similar study using neutron reflection (Shepherd et al., 2022)"

391: I think it would be helpful here or elsewhere to stress that the products of oxidation are taken to be volatile, since film thickness is used to follow the reaction.

Corrected: The text now reads: "It is unclear whether the residual film is an unreactive portion of the original film or an oxidised unreactive product (or even a mixture of both). Oxidised material that has left the interface has either dissolved in the aqueous sub-phase or is volatile and entered the gas-phase". The thickness is not used to follow the reaction, the product  $\rho\delta$ , is used to follow the progress of reaction and is equal to the scattering length per unit area at the air-water interface as explained on line 172-173. "For the atmospheric aerosol extracts films exposed to gas-phase OH radicals, scattering length density per unit area,  $\rho\delta$ , was plotted as a function of time (Jones et al., 2017) instead "

392: The timeseries of uptake coefficients, e.g., approaching zero, can also be mentioned here.

The following text has been added "...and the calculated uptake coefficients approach zero".

Citation: https://doi.org/10.5194/egusphere-2024-2367-RC2

---

## Author Response (AR2)

**The lifetimes and potential change in planetary albedo owing to the oxidation of organic films extracted from atmospheric aerosol by hydroyxl (OH) radical oxidation at the air-water interface of aerosol particles**

Rosalie Shepherd, Martin King *, Andrew Ward, Edward Stuckey, Rebecca Welbourn, Neil Brough, Adam Milsom, Christian Pfrang, and Thomas Arnold

We thank the reviewer, and the editor for their time and comments, which have helped improve this paper.

I consider the majority of the comments settled in the revised version, except the point of representativeness of the conditions of the insoluble extracted organics in the trough for the conditions under which the same components were exposed to the surrounding gas phase while in ambient air.

Both reviewers seemed to understand that spreading the extracted component on the water surface in the trough leads to a monolayer. However, in the particles, from which these components were extracted, they may not have necessarily formed a monolayer but may have been part of a complex morphology. This would mean that they may have a different lifetime with respect to reaction with OH or also with respect to their contribution to the optical properties. It seems that the present experiments are certainly representing the situation when ambient particles are activated into cloud droplets, so that these insoluble surfactants can spread on the surface of the droplets.

I suggest that a caveat about this is included in the discussion and conclusions, and the tonality of the abstract adapted to this.

The following text has been added: "Although Fig 1 demonstrates that the material extracted from atmospheric aerosol produces a stable thin film at the air-water interface and thus may form these films on aqueous droplets in the atmosphere, it does not necessarily imply that the aerosol from which these insoluble surfactants were extracted had such a monolayer and may have had a more complex morphology."

An additional point helping to clarify would be to mention the mass fraction of the material spread in the trough of the total particle mass from which it was extracted.

The mass fraction of the spreading solution is not precisely known as enough material is added to achieve the required surface pressure, but an estimate was included in the last revision and identifies the mass per unit volume as approximately $7 \times 10^{-5}\ g\ ml^{-1}$ with the following text in line 150: "A similar surface pressure produced by either a monolayer of fatty acid molecules like oleic acid would give a surface coverage of $\sim 2 \times 10^{18}$ molecule $cm^{-2}$ , (King et al., 2010), about 16 μg of material, or by using the mass density of the spreading solution ($\sim 7 \times 10^{-5}\ g\ ml^{-1}$ ) of similar aerosol extract (Shepherd et al., 2018) to those used here demonstrates about 10μg of materials was added to the trough. "

The following changes were also made: "TEXT" removed from optional copyright statement. Crossing red and green plots changed on Figures 1,2 and 6. Figure 3, the marker shape is changed as well as the colour. Figure 6 has different marker shapes as well as colours.

---

## Author Response (AR3)

**The lifetimes and potential change in planetary albedo owing to the oxidation of organic films extracted from atmospheric aerosol by hydroyxl (OH) radical oxidation at the air-water interface of aerosol particles**

Rosalie Shepherd, Martin King *, Andrew Ward, Edward Stuckey, Rebecca Welbourn, Neil Brough, Adam Milsom, Christian Pfrang, and Thomas Arnold

We thank the editor for their time and comments, which have helped improve this paper. The suggested text from the editor was very helpful.

Thank you for sending the revised version of your manuscript. Although you have included requested discussion point in the revised manuscript, I feel that the abstract and conclusions can remain ambiguous as to which scenario the results apply to.

As you indicated, the spread aerosol extract likely stems from millions of individual aerosol particles. If the organic materials in the individual particles are not partitioning to the air-water interface, then the lifetime with respect to OH and the optical properties of the aerosol particle and cloud droplet will be different from what is reported. Reading the abstract and conclusion the audience of ACP may be inclined to take the results as generally valid (i.e., a thin film always establishes with measured properties). However, strictly speaking this is only the case if the organic materials in the aqueous aerosol particle and droplet partition to the air-water interface. In this regard the reviewer is correct in pointing this out.

I think minor adjustments can resolve this ambiguity without compromising the results of your study. For example, in abstract on line 5 you could state "Assuming the aerosol extracts reflect thin films on aqueous particles and cloud droplets, modelling....". In conclusions you could state "...aqueous core-shell systems.", and "...depending on the ambient hydroxyl radical concentration assuming aqueous particles and cloud droplets".
Obviously, I will leave it to you to make these minor changes to increase clarification.

We have adopted the Editors very useful corrections and the abstract now has the following text:-

"Assuming the material extracted from atmospheric aerosol produces thin films on aqueous particles and cloud droplets, modelling the oxidation kinetics with KM SUB suggests half-lives of minutes to an hour..."

And the conclusions now has the following text:-

"Optimised KM SUB kinetic models suggested that the atmospheric lifetime of the reactive component of the films studied can vary between minutes and days depending on the ambient hydroxyl radical concentration assuming aqueous particles and cloud droplets."